# Application of Ultrashort Lasers in Developmental Biology: A Review

**Inna V. Ilina ***[ID] **and Dmitry S. Sitnikov** [ID]

Joint Institute for High Temperatures of the Russian Academy of Sciences, Izhorskaya st. 13, Bd. 2, 125412 Moscow, Russia
* Correspondence: ilyina_inna@mail.ru

**Abstract:** The evolution of laser technologies and the invention of ultrashort laser pulses have resulted in a sharp jump in laser applications in life sciences. Developmental biology is no exception. The unique ability of ultrashort laser pulses to deposit energy into a microscopic volume in the bulk of transparent material without disrupting the surrounding tissues makes ultrashort lasers a versatile tool for precise microsurgery of cells and subcellular components within structurally complex and fragile specimens like embryos as well as for high-resolution imaging of embryonic processes and developmental mechanisms. Here, we present an overview of recent applications of ultrashort lasers in developmental biology, including techniques of noncontact laser-assisted microsurgery of preimplantation mammalian embryos for oocyte/blastomere enucleation and embryonic cell fusion, as well as techniques of optical transfection and injection for targeted delivery of biomolecules into living embryos and laser-mediated microsurgery of externally developing embryos. Possible applications of ultrashort laser pulses for use in Assisted Reproductive Technologies are also highlighted. Moreover, we discuss various nonlinear optical microscopy techniques (two-photon excited fluorescence, second and third harmonic generation, and coherent Raman scattering) and their application for label-free non-invasive imaging of embryos in their unperturbed state or post-laser-induced modifications.

**Keywords:** ultrashort laser pulses; femtosecond laser; laser microsurgery; embryo development; oocyte; nonlinear optical microscopy; cell fusion; enucleation; optoinjection

## 1. Introduction

Lasers have become a powerful tool in basic biological research and medicine [1]. A massive variety of laser applications exists in ophthalmology [2], dermatology [3], and surgery [4,5]. The most fascinating and promising implementations of laser technology are based on the ability of laser light to perform precise micromanipulations (microdissections) at the cellular and even subcellular levels. Recently, some valuable laser-based devices and technologies have been developed. For example, laser-assisted microdissection and pressure catapulting by the PALM Microbeam system (Carl Zeiss, Oberkochen, Germany) allow fast and noncontact isolation of the desired cell segment from the undesired one [6,7]. The technology is widely applied in the molecular analysis of cancer specimens, enabling it to overcome one of the major challenges related to tissue heterogeneity. Laser-based technologies are also promising approaches in tissue engineering and regenerative medicine. The technology of laser guidance direct writing proposed by Odde and Renn [8] in 1999 utilized laser-induced optical forces to guide and deposit particles of various sizes onto solid surfaces. Many developments and novel techniques of laser-assisted bioprinting have been proposed [9] since then, making possible the construction of various tissues or organs such as vessels, heart, and liver [10].

Berns et al. [11] in 1981 and others [12,13] have repeatedly highlighted the benefits of using lasers in cell and developmental biology. As laser technology has made great progress over the past 40 years, the range of possible applications of lasers in developmental biology has also substantially increased.

A key achievement in laser physics is the invention of ultrashort lasers. The physicists Gérard Mourou and Donna Strickland, the winners of the 2018 Nobel prize, made a breakthrough in high-intensity, ultra-short optical pulse generation with their chirped-pulse amplification technique. Ultrashort laser pulse (ULP) commonly implies a pulse with duration in the pico- (1 ps = $10^{-12}$ s) or femtosecond range (10 fs = $10^{-14}$ s) and high peak intensity. Shorter pulse durations are hard to obtain and implement due to light dispersion in the matter it passes through along its way from the laser to the sample, which has to be taken into account and compensated. Laser pulses with durations shorter than 80–100 fs are commonly not applied. Due to high intensity of laser pulses new mechanisms of laser-matter interaction take place. Nonlinear optical effects in the focal volume of tightly focused ULP offer several advantages compared to long-pulsed (nano- and microsecond) or continuous wave (CW) lasers. The absence of adverse effects like out-of-focus light absorption or substantive heat transfer from the focal point to the surrounding media facilitates high precision microsurgery of cells and tissue with minimal collateral damage. Using ULPs has also created new possibilities for imaging cells and tissues, as nonlinear optical microscopy offers several advantages over conventional microscopies, like high resolution and non-invasiveness without needing exogenous markers.

This review presents an overview of recent applications of ultrashort lasers in developmental biology, including techniques of noncontact laser-assisted microsurgery of preimplantation mammalian embryos for oocyte/blastomere enucleation and embryonic cell fusion. Laser-mediated microsurgery of externally developing embryos aimed at studying cell mechanics and microscopic fluid flows or wound healing post localized damaging of embryonic tissues is presented. Laser-based techniques to deliver exogenous material substances into embryonic cells, optoinjection, and laser transfection are also discussed. Various techniques of nonlinear optical microscopy for the precise imaging of embryo structures, as well as recent advances in simultaneous application of nonlinear optical microscopy and laser microsurgery, are presented. Possible applications of ultrashort laser pulses for laser microsurgery and nonlinear optical microscopy of embryos in assisted reproduction technologies are discussed.

## 2. Interaction of Ultrashort Laser Pulses with Matter

Let us consider the peculiarities of ULPs interaction with matter to see all the advantages provided. The nonlinear absorption process, a distinguishing feature of ULP, is briefly described below. Water is commonly used as a basis for modeling processes during ULPs-tissue interaction because water has similar properties to biological media in absorptance, optical breakdown threshold, and thermal properties [14,15]. Since 1991, water has been considered an "amorphous semiconductor" [16] with a band gap (BG) of 6.5 eV separating the valence band (VB) and the conduction band (CB). The complex energy structure of water is thoroughly studied; modern concepts can be found elsewhere [17,18].

Electrons are the key players in the absorption process of photons of the laser pulse. The acquired energy results in electron transition from the VB to the CB. However, water's bandgap energy is higher than a photon's energy in the visible or near-infrared (NIR) light region. Being focused, ULPs are characterized by high intensity, i.e., high concentration of photons, making photoionization possible (Figure 1). The photoionization can typically occur through two processes called multiphoton absorption [19] or tunneling ionization [20]. Each process's probability depends on the field strength and frequency of the electromagnetic field. These first free electrons in the CB during collisions with ions or atomic nuclei facilitate the process of further photon absorption from the incoming radiation (the so-called inverse Bremsstrahlung absorption). The electrons accumulate the kinetic energy, help transfer the remaining electrons from the VB to the CB through impact and avalanche ionization (see, e.g., Ref. [21] for details), and form the so-called low-density plasma in the laser beam focus area.

The interaction between laser radiation and water or biological tissue is a complex process where particular laser-induced effects depend on several values and parameters

like laser power (intensity or pulse energy), wavelength, pulse duration, repetition rate, and characteristic properties of the tissue to be laser-processed.

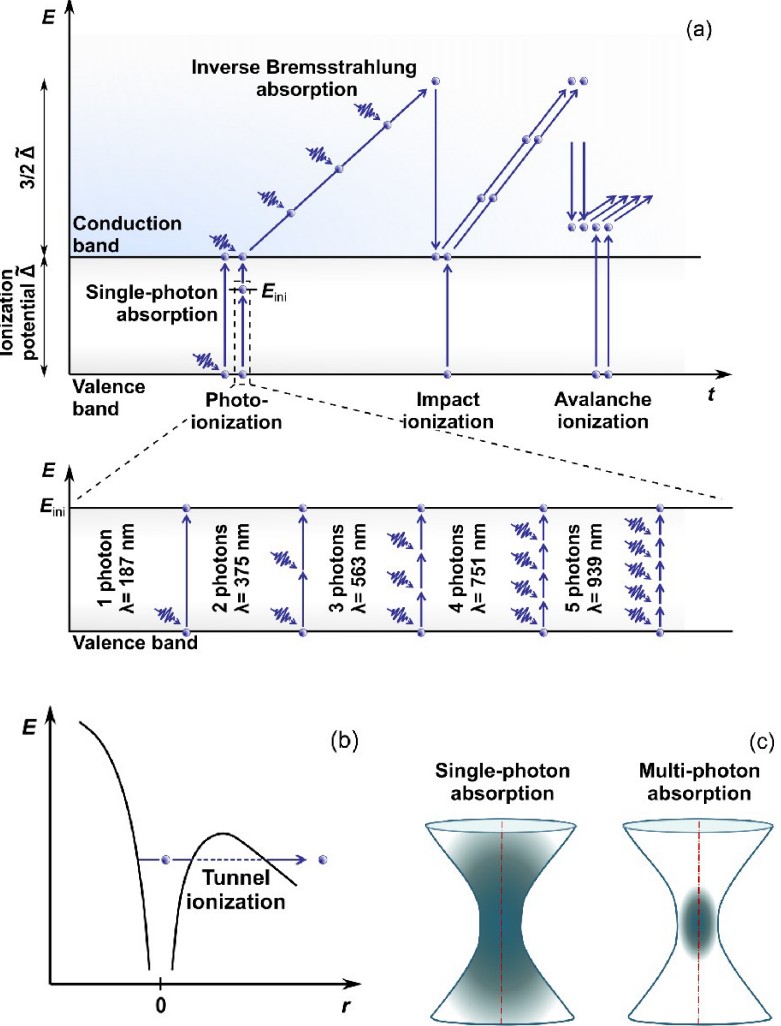

**Figure 1.** Water ionization scheme for laser energy deposition: (**a**) multiphoton, avalanche and impact ionization, (**b**) tunnel ionization, and (**c**) Single- and multiphoton absorption geometry. Adapted from open-access (CC BY license) source, Ref. [22].

Formation of electron plasma in laser focus results in temperature increase. A detailed description of absorption processes and temperature estimation can be found in a series of studies by Vogel et al. [23,24]. The authors also demonstrated [25] that a temperature of 100 °C can easily be reached in laser focus (wavelength $\lambda$ = 800 nm, pulse duration $\tau$ = 170 fs, and repetition rate $f_{rep}$ = 80 MHz) at intensity $I_{peak}$ = 3.3 × 10$^{12}$ W cm$^{-2}$ causing photothermal effect of denaturation of biomolecules. This effect is also accompanied by free-electron-induced chemical effects due to the high reactivity of the electrons.

The chemical effects in biological media are commonly divided into two groups: (1) Changes in the water molecules create reactive oxygen species (ROS), subsequently affecting organic molecules. OH* and $H_2O_2$ oxygen species have been shown to cause cell damage [26]; the process of their formation following the ionization and dissociation of water molecules is described in detail [27]. (2) Direct changes of the organic molecules are about the capture of electrons into an antibonding molecular orbital, initiating the biomolecules' fragmentation [23,26,28]. For multiple pulses, this accumulative effect can lead to dissociation/dissection of biological structures exposed to low-density plasmas generated by femtosecond laser radiation (e.g., DNA strand breaks [28]). It has recently

been shown that femtosecond laser pulses with relatively small peak intensity ($\lambda$ = 794 nm, $\tau$ = 100 fs, and $f_{rep}$ = 80 MHz, $I_{peak} \leq 4 \times 10^{11}$ W cm$^{-2}$) could act as a highly localized ionizing tool [29]. They can induce complex DNA damage involving different repair pathways. Thus, the local ionizing effect should be considered when femtosecond laser radiation is used for biomedical applications.

Another type of effect induced by ULPs is the tensile stress in the medium. Local temperature increase results in thermal expansion. Because energy delivery by ULP is faster than the medium can expand [30], substantial transient stresses are developed. In water, when the tensile strength of the liquid is exceeded, it causes the formation of a cavitation bubble. Thus, the tensile stress wave may induce object fracture even after a temperature rise is too small to produce thermal damage [31].

The multiphoton absorption process has several advantages, dealing with transparent materials not absorbing at low intensities at a given wavelength. Applying ULPs with high photon concentration increases the probability of multiphoton absorption at high intensities in laser focus. Thus, firstly, laser-matter interaction in a transparent medium occurs in a small volume only in the vicinity of the beam waist (Figure 1c), thus enabling selective laser-based microsurgery of cells or subcellular structures (e.g., embryo cells or intracellular organelles) without disrupting surrounding tissues or cell membranes. Secondly, nonlinear absorption prevents the medium from heating along the laser beam path. Furthermore, the spatial distribution of the free electrons in focal volume is narrower than intensity distribution by a factor of $\sqrt{k}$, where $k$ is the number of photons captured simultaneously by the electron (Figure 2). Thus, low-density plasma's heating, thermomechanical, and chemical effects produced are very well localized in a small volume, making possible microsurgery with subdiffraction spatial resolution. Moreover, laser surgery with ultrashort lasers can be conducted with much less average power than long-pulsed or CW lasers. Application of the latter for surgery usually requires staining target structures (to create additional energy levels within the BG) unless laser powers higher than 1 W are used. The average power required for femtosecond laser-based microsurgery was substantively lower (cf. Cell microsurgery utilizing CW argon laser at wavelengths $\lambda$ = 488 nm/514 nm required laser power of 1 W [32], while an average power of 30 mW [33,34] was enough to perform microsurgery with femtosecond Ti:sapphire laser at a wavelength $\lambda$ = 800 nm).

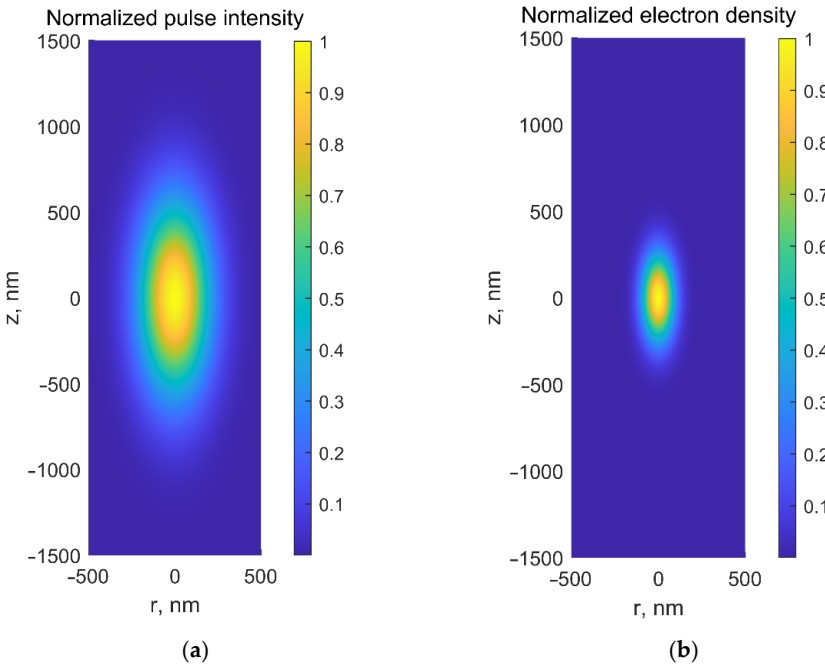

**Figure 2.** (**a**) Normalized laser intensity and (**b**) normalized electron-density distributions in the focal area at $\lambda$ = 800 nm for NA = 1.3.

When considering ULPs, two types of femtosecond lasers should be mentioned, with repetition rates in the MHz and kHz frequency ranges. The former source comprises typical seed oscillators, generating pulse trains with small energies of tens of nanojoules and periods of about 10 ns. The latter source is additionally equipped with a regenerative amplifier, enabling an increase of pulse energy up to the millijoule level for the reduced pulse repetition rate. Thus, two different microsurgery scenarios should be considered. Due to low pulse energy, accumulative chemical effects in a low-density plasma regime justify microsurgery in case of MHz pulse repetition rate, while laser microdissection at kHz repetition rate mainly relies on thermoelastically induced formation of transient cavities.

## 3. Application of Ultrashort Laser Pulses for Nonlinear Microscopy of Embryos

Traditional single-photon fluorescence (SPF) microscopy covers the radiation of light-emitting probes (fluorescent proteins, dye molecules, and semiconductor nanoparticles) chemically associated with specific biological elements (proteins, DNA, and phospholipids). Probes' spatial distribution is recorded in widefield or point scanning mode with subsequent image reconstruction (Figure 3). Additional equipment like high numerical aperture lenses and pinholes is required to achieve spatial filtering and remove out-of-focus light or glare. Studying the dynamics of embryonic development using SPF can be impeded as cell exposure to visible [35–37] or high-intensity light [38] can cause oocyte or embryo damage. Moreover, using classic SPF microscopy is difficult as a need exists to monitor embryo development for a comparatively long period (from several hours to several days). This fact highlights the importance of improving long-term fluorescence imaging techniques.

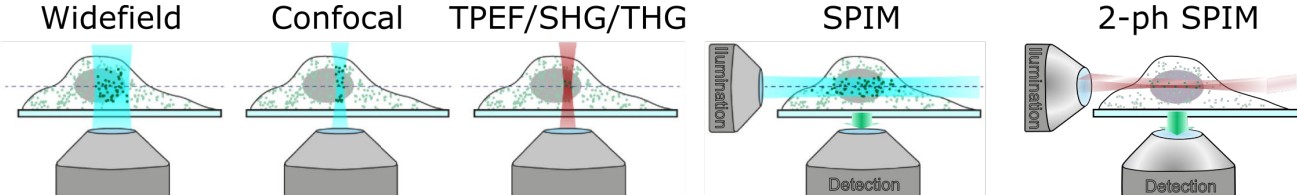

**Figure 3.** Principal schemes for microscopy modalities. Adapted from an open-access (CC BY license) source; Ref. [39].

The development of nonlinear optical (NLO) microscopy overcomes most previous limitations, enabling effective imaging at greater depths with higher spatial resolution without staining. The most suitable for biological research NLO-microscopy modalities include two-photon excited fluorescence (TPEF) [40], generation of the second and third harmonics (SHG and THG) [41,42], and coherent Raman scattering (CRS) microscopy [43].

Application of ULPs in NLO microscopy revealed some features related to pulse transportation from laser to the sample. While passing though microscope optics, the low frequencies of broadband pulse spectrum travel faster than high frequencies due to group velocity dispersion effect. Thus, original short pulse becomes longer and demonstrates time dependence of its instantaneous frequency (chirp phenomenon). This results in decrease in pulse intensity, a key parameter for multiphoton absorption. To deliver laser pulse with duration of several tens of femtoseconds to the sample a special tunable-chirp laser systems have to be applied. They allow to generate a "negative-chirp" pulse (at laser output), high frequencies of which travel in the front of the pulse. The value of this negative dispersion is chosen to be compensated by positive dispersion in microscope optics.

Two-photon fluorescence microscopy, also known as two-photon laser scanning (TPLS) microscopy, is based on the simultaneous absorption of two photons. This modality allows for the visualization of both exogenous and endogenous fluorophores [44]. Because TPEF is based on the nonlinear phenomenon, it offers a higher penetration depth of illumination, higher spatial resolution, and actual 3D scanning capability. Unfortunately, photobleaching and photodamage may occur at the focal volume where photochemical interactions occur.

The SHG is a second-order nonlinear process, in which two photons interacting with a nonlinear material are upconverted to form a new photon with twice the frequency of initial photons. The medium should be noncentrosymmetric to obtain the SHG signal. It is generated in collagen and astroglial fibers, myofilaments, and polarized tubulin assemblies like mitotic spindles. The SHG process originates from induced polarization instead of actual absorption, resulting in a substantial decrease in the probability of phototoxicity and photobleaching. SHG microscopy is considered relatively non-invasive, since there is no need in adding exogenous markers. Besides, there is no energy contribution to the biological object, since the energy of a pair of excitation photons and that of the SHG photon are equal.

In contrast to SHG, THG microscopy does not require asymmetry of molecules to generate the signal. The third-harmonic waves generated before and after the laser focal point interfere destructively, resulting in zero net THG [45] if the medium around the focal point is homogeneous. In inhomogeneity, like an interface between two media and a mismatch of refractive indices [46,47], the symmetry along the optical axis is broken, and the third harmonic wave can be detected. In cells, THG signals are commonly detected from mitochondria and lipid bodies. Higher harmonic-generation microscopy, including SHG and THG, leaves no energy deposition to the interacted matters providing a truly non-invasive modality. Details on these techniques, seeming ideal for in vivo imaging of live specimens without any preparation, can be found elsewhere [48].

Raman scattering is a molecule/material identification technique based on the characteristic vibrational spectrum. In CRS microscopy, the Raman signal is generated from a coherent superposition of the molecules in the sample. The sample is irradiated by two synchronized ULPs of different frequencies, the pump $\omega_p$, and the Stokes $\omega_s$. When their difference, $\Omega = \omega_p - \omega_s$, matches the vibrational frequency, resonant excitation and in-phase vibration of all the molecules in the focal volume are observed. Two most widely applied CRS techniques are the stimulated Raman scattering (SRS) [49] and the coherent anti-Stokes Raman scattering (CARS) [50] (see the diagrams for the aforementioned microscopy modalities in Figure 4). Parodi et al. have compared and described these techniques in detail [44]. Wang et al. present recent advances in the development of label-free optical imaging techniques for application in developmental biology [51].

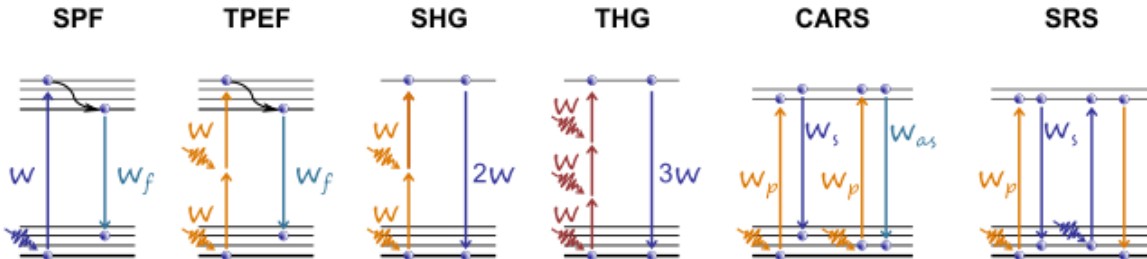

**Figure 4.** Energy level diagrams for various microscopy modalities (from left to right) for single-photon (SPF) and two-photon excited (TPEF) fluorescence, second (SHG) and third (THG) harmonic generation, coherent anti-Stokes Raman scattering (CARS), and stimulated Raman scattering (SRS). Adapted from an open-access (CC BY license) source; Ref. [22].

Another microscopy modality that should be mentioned is light-sheet microscopy, a century-old known as selective-plane illumination microscopy (SPIM). The sample is illuminated with a plane of visible light, generating fluorescence from a thin optical section, which is then imaged with a widefield camera orthogonally to the light sheet. The orthogonal geometry between the illumination and detection pathways, enables higher imaging speed owing to the parallel image collection and reduces photodamage because only a single focal plane of the sample is illuminated at a time [52]. This microscopy modality has been upgraded in several studies [52,53] by applying ULPs combined with TPLS. Such a scheme, called 2-ph SPIM, combines the advantages of both modalities. TPLS provides

high penetration length in scattering tissues, while conventional SPIM (or 1-ph SPIM) offers higher acquisition speed. Near-infrared ULPs are used to create a two-photon excitation light sheet, which performs axial sectioning. The 2-ph SPIM also offers better axial resolution than 1-ph SPIM at large sample depth. Firstly, scattering of excitation light is reduced at near-infrared (compared to visible) wavelengths, providing better preservation of the light-sheet thickness. Secondly, quadratic dependence on the excitation light intensity of two-photon–excited fluorescence makes scattered illumination light less important, as fluorophore excitation is spatially confined to only the highest intensity part of the beam, thus preserving axial resolution even when the light sheet is thickened by scattering [52]. Bidirectional illumination can be applied to increase the useful field of view of camera.

Many studies have been published over the last decades reporting on applying nonlinear microscopy in developmental biology. As several comprehensive reviews on this topic have already been published, we mainly discuss studies not covered earlier in this section.

*Drosophila melanogaster* and zebrafish (*Danio rerio*) are popular model organisms. The former is considered to be a powerful platform for understanding the interplay between genetics and biophysics (see the review [54] and references therein) and for reverse-engineering multicellular systems. *Drosophila* and zebrafish require minimal preparations and maintenance for live imaging. In contrast, mammalian models, such as mice, require optimal embryo culture for live imaging [55]. The mouse is the only mammalian organism with well-established genetic engineering strategies to generate various disease model. The efficient development and use of these mutant mice necessitate phenotypic imaging analyses of the created models, accentuating the importance developing efficient approaches for high-resolution mouse embryonic imaging [51]. A recent review discusses studies employing TPEF, SHG, THG, or CRS microscopy techniques for mouse embryo quality assessment and strategies for embryo selection with high implantation potential [22]. Moreover, SHG microscopy can be a valuable imaging tool for exploring mouse embryonic cardiogenesis and biomechanics [56]. SHG combined with TPEF microscopy of genetic mouse model enabled visualization of establishing cardiac fibers and resulted in detection of a substantial increase in fibrillar content and organization during the first 24 h after initiation of contractions.

Zebrafish is an excellent animal model for in vivo studies due to its transparency during embryogenesis, its amenability to optical imaging, and the easiness of transgenic line generations [57]. Zebrafish embryos are also ideal model vertebrates for high-throughput toxicity [58]. Detailed study of live zebrafish embryos and larvae using non-invasive TPEF, SHG, and Light-Sheet microscopy techniques can be found in a recent review (see Ref. [57] and references therein). The review [57] covers the pros and cons of the listed modalities and their application to study, for example, the dynamics of cytological construction, development of cranial neurons and blood vessels during embryogenesis, and organization of collagen fibers during the fin wound healing.

Higher-harmonic generation techniques (SHG and THG) have a solid potential for long-term in vivo study of the nervous system, including genetic disorders, axon pathfinding, neural regeneration, neural repair, and neural stem cell development [59]. By utilizing endogenous SHG as the contrast of polarized nerve fibers and THG to reveal morphological changes, the vertebrate embryonic nervous development was successfully observed in a live zebrafish embryo from the very beginning using a near-infrared light source (Cr: forsterite laser, $f_{rep}$ = 110 MHz, $\lambda$ = 1230 nm, $\tau$ = 140 fs, and $P_{av}$ = 100 mW) [59]. Generation of SHG from myelinated nerve fibers and the outer segment of the photoreceptors with a stacked membrane structure were also reported for the first time.

SHG microscopy has also been used to explore the trachea system, developing muscle structures in 2nd-instar larva, and the lipid bodies in *Drosophila* cells [60,61] to investigate the structure of *Drosophila* sarcomeres and to visualize myocyte activity in terms of rhythmic muscle contraction of both larval and adult stages [62,63]. In a later study [64], a question of laser-induced toxicity of THG imaging has been raised. The influence of imaging rate, wavelength, and pulse duration on the short-term and long-term perturbation of *Drosophila*

embryos development has been studied and the criteria for safe imaging have been defined. Conducted studies enabled authors to derive general guidelines for improving the signal-to-damage ratio in two-photon (TPEF/SHG) or THG imaging (see examples in Figure 5). Imaging of lipid droplets that exhibit highly dynamic behavior in early *Drosophila* embryo provides a robust platform for the investigation of shuttling by kinesin and dynein motors. Real-time imaging and quantification of droplet motion using an efficient and convenient technique termed "femtosecond-stimulated Raman loss" microscopy (Figure 5f–h) enabled Dou et al. [65] to develop a velocity-jump model to predict the population distributions of droplet density.

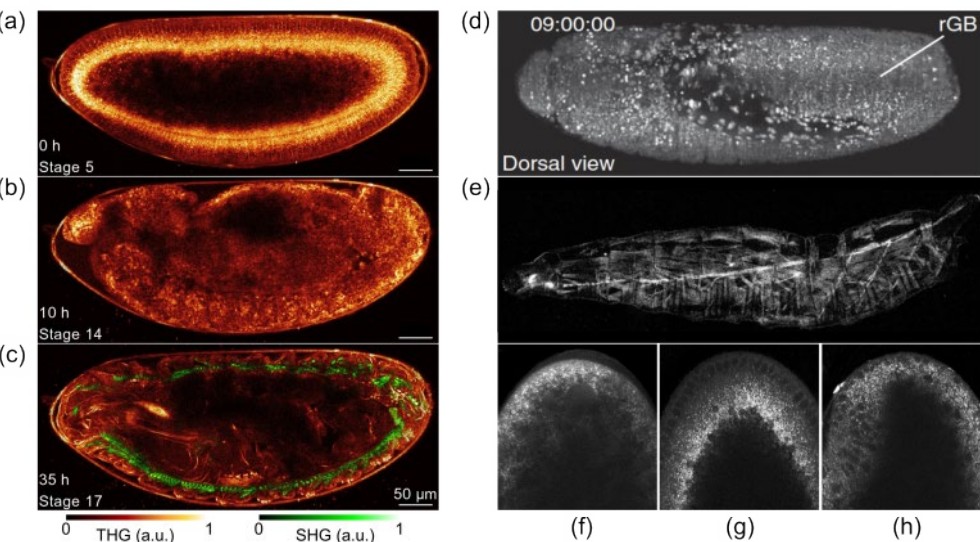

**Figure 5.** (**a–c**) Long-term THG-SHG imaging of *Drosophila* embryonic development. 2D THG-SHG imaging at a wavelength of 1180 nm of a wild-type *Drosophila* embryo during 36 h starting from stage 5 up to the larvae stage (i.e., until hatching). Reprinted from an open-access (CC BY license) Ref. [64]. (**d**) Combination of TPEF with SPIM that delivers high imaging speeds and near complete physical coverage of the embryo while reducing photobleaching and phototoxic effects. (**e**) SHG microscopy visualizes muscular architecture and trachea system in detail without fluorophore labeling. Adapted from an open-access (CC BY license) Ref. [54]. (**f–h**) fSRL (i.e., femtosecond stimulated Raman loss) images of lipid droplet global distribution during *Drosophila* early embryogenesis at phases of nuclear cycle 13, midcellularization, and gastrulation, respectively. Adapted from an open-archive source (Ref. [65]).

Recent technological advances have facilitated whole-brain recording in small organisms, including *Drosophila melanogaster*, nematode *Caenorhabditis elegans*, and zebrafish *Danio rerio* [66]. *Drosophila melanogaster* is a popular model for brain studies due to a small number of neurons interacting in limited circuits, allowing analysis of individual computations or steps of neural processing. Light sheet fluorescence microscopy also allows automated imaging of *Drosophila melanogaster* embryos [67]. Memeo et al. have developed a new microscope on a chip comprising optical and fluidic components (for embryo alignment) [67]. Two different *Drosophila* populations expressing GFP and mRFP have been successfully processed at two wavelengths (488 nm and 561 nm, correspondingly), and 3D observation of the internal organs, segmentation, and quantification of their volume has been demonstrated.

Applying THG microscopy makes it possible to detect altered photoreceptor development in *Drosophila* pupal eye that may be helpful in clinically relevant conditions associated with photoreceptor degeneration [68]. In their later study, Karunendiran et al. [69] successfully applied polarimetric SHG microscopy to characterize the changes in myosin accumulation in the *Drosophila* larva body wall muscles. Experiments revealed changes in somatic muscle striated patterns and reduced signal intensity correlated with diminished

order of myosin filaments. Polarization-resolved SHG technique enabled extracting the nonlinear susceptibility tensor components ratio of myosin fibrils in the body wall muscles of *Drosophila* larva [70].

### 4. Oocyte/Blastomere Enucleation and Embryonic Cell Fusion by Application of Ultrashort Laser Pulses

This section discusses a selection of ultrashort laser-based techniques applied in fundamental research for embryo manipulation, including laser-induced embryonic cell enucleation and fusion. Table 1 summarizes the main results and laser exposure parameters.

**Table 1.** Ultrashort laser-induced enucleation and fusion of oocytes/blastomeres.

| Type of Oocytes/ Embryos | Type of Manipulation | Laser Exposure Parameters | Efficiency of Enucleation/Fusion | References |
|---|---|---|---|---|
| Mouse | Functional enucleation (nucleoli, cytoplasm, metaphase plate, pronucleus irradiation), Blastomere fusion | Ti:sapphire laser $\tau = 2$ ps, $\lambda = 800$ nm, $f_{rep} = 80$ MHz, $P_{av} = 0.4$ W, $E = 5$ nJ, t = 0.3 s | Efficiency of nuclear inactivation of one of two blastomeres: 75–80% Efficiency of oocyte inactivation after metaphase plate irradiation: 100% Fusion efficiency/Blastocyst formation: 35.8%/33.3%–66.7%/95% (dependent on the site of laser impact on the blastomere contact border) | [71] |
| Porcine | Functional enucleation | Ti:sapphire laser $\tau \sim 275$ fs in the sample, $\lambda = 720$ nm, $f_{rep} = 1$ MHz, $E = 2.5$ nJ, $\upsilon = 100$ μm s$^{-1}$ | Efficiency of enucleation: 96% | [72] |
| Porcine | Blastomere fusion | Ti:sapphire laser $\tau \sim 275$ fs in the sample, $\lambda = 720$ nm, $f_{rep} = 80$ MHz, t = 20 ms, F = 0.36 J cm$^{-2}$ ($P_{av} = 430$ mW), t = 100 ms, F = 0.23 J cm$^{-2}$ ($P_{av} = 670$ mW) | Fusion efficiency/Cell viability/Blastocyst formation: 54%/95%/70% (for t = 20 ms) 44%/73%/43% (for t = 100 ms) | [73] |
| Mouse | Blastomere fusion | Cr:forsterite seed oscillator and regenerative amplifier $\tau = 100$ fs, $\lambda = 620$ nm, $f_{rep} = 10$ Hz, $E = 30$–50 nJ | Fusion efficiency/Blastocyst formation: 89%/50% | [74,75] |
| Mouse | Blastomere fusion | Ti:sapphire laser $\tau = 100$ fs (at the sample), $\lambda = 800$ nm, $f_{rep} = 80$ MHz, $P_{av} = 80$ mW, E = 0.3, 1, 2 nJ, t = 15, 30, 60 ms, $I_{peak} = 2.0$–$13.2 \times 10^{11}$ W cm$^{-2}$ | Fusion efficiency/Blastocyst formation rate: 29%/49% (for E = 1 nJ, t = 30 ms) | [76] |
| Mouse | Oocyte fusion, fusion of oocyte with blastomere, fusion of 2–3 blastomeres inside 4-cell embryos | Ti:sapphire laser $\tau = 2$ ps, $\lambda = 800$ nm, $f_{rep} = 80$ MHz, $P_{av} = 0.4$ W | Oocyte fusion efficiency: 46/7% Oocyte/blastomere fusion efficiency: 21.2% Efficiency of two-blastomere fusion inside 4-cell embryo/Blastocyst formation: 61.5%/78.1% Efficiency of two pairs of blastocyst fusion inside 4-cell embryo/Blastocyst formation: 52.2%/90% Efficiency of three blastomere fusion inside 4-cell embryo/Blastocyst formation: 44.4%/50% | [77] |

Table 1. *Cont.*

| Type of Oocytes/ Embryos | Type of Manipulation | Laser Exposure Parameters | Efficiency of Enucleation/Fusion | References |
|---|---|---|---|---|
| Mouse | Oocyte fusion, Blastomere fusion | Ti:sapphire laser $\tau = 100$ fs in the sample, $\lambda = 690–1000$ nm, $f_{rep} = 80$ MHz, $P_{av} = 0.8$ W (before the objective), $E = 1$ nJ, t = 15 ms (for oocyte fusion), t = 30 ms (for blastomere fusion) | No data | [78] |

Femtosecond lasers can be used to perform the main steps of somatic cell nuclear transfer (SCNT). One of the key steps in SCNT is called enucleation (Figure 3a), followed by the transfer and fusion of a somatic cell into an enucleated oocyte [79]. Applying femtosecond lasers has substantial benefits over conventional approaches such as electrofusion and microinjection with a piezo-manipulator and may improve the efficiency of somatic cell clone production [72]. The interaction of ULP with biological tissues is based on nonlinear absorption, resulting in higher penetration depths and absence of out-of-focus absorption and hence reduced (or no) damage to the surrounding structures and organelles.

When femtosecond laser pulses are applied for metaphase plate ablation, a so-called functional enucleation (step 3* in Figure 6a) is performed instead of removal of the haploid chromosomes comprising the meiotic spindle complex from a metaphase II (MII)-stage oocyte (step 3 in Figure 6a). During the process of laser-based functional enucleation, DNA inactivation and parthenogenetic developmental arrest occur, leaving the other organelles and the cytoplasm intact. Figure 6b demonstrates the metaphase II oocyte with clearly visible metaphase plate stained by Hoechst 33342 dye. The metaphase plate was partially destroyed in Figure 6c and fully destroyed in Figure 6d by application of femtosecond laser pulses (trains of pulses with 100 fs duration, 80 MHz repetition rate, 0.5 nJ pulse energy and pulse train duration 60 ms) [80].

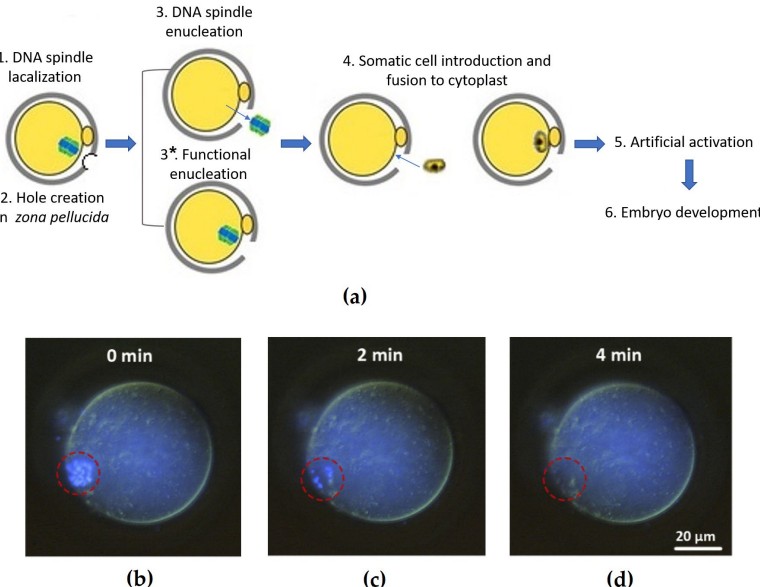

**Figure 6.** (**a**) Schematic representation of the main steps of somatic cell nuclear transfer (mechanical spindle extraction (step 3) or functional enucletion with laser can be performed (step 3*)) and (**b–d**) the process of metaphase II oocyte enucleation by femtosecond laser pulses (Adapted from open-access source, Ref. [80]).

Picosecond laser pulses ($\tau$ = 2 ps, $\lambda$ = 800 nm, and $f_{rep}$ = 80 MHz) were applied for efficient functional enucleation of mouse oocytes and blastomeres [71] and femtosecond ones ($\tau$ = 140 fs/275 fs in the sample, $\lambda$ = 720 nm, $f_{rep}$ = 1 MHz, $E$ = 2.5 nJ, $\upsilon$ = 100 $\mu$m s$^{-1}$) for inactivating porcine oocytes [72]. Karmenyan et al. [71] reported the optimal conditions for nucleus inactivation in ovulated MII oocytes, activated oocytes, and embryos containing pronuclei and nuclei and demonstrated the efficiency of laser-based nucleus inactivation without reducing the cytoplasmic volume. While Karmenyan et al. [71] visualized the metaphase plate in mouse oocytes before inactivation by Hoechst staining in combination with UV illumination, Kuetermeyer et al. [72] used the very same femtosecond laser system for both three-dimensional (3D) imaging and nuclear inactivation. Using femtosecond laser pulses with NIR wavelengths for metaphase plate imaging improved oocyte viability and developmental potential compared to UV illumination, which may cause damage to the oocyte cytoplasm [81]. The efficiency of functional inactivation reached 85% [71] and 96% [72] with maintaining intact morphology over a long period. None of the enucleated oocytes underwent cleavage and continued parthenogenetic development. It should be mentioned that ablation with suitable irradiation doses in the cytoplasm instead of the metaphase plate did not cause the arrest of embryo development. Such embryos continued to develop with no observable difference from the control embryos. The potential for automation of the enucleation procedure has been shown by combining multiphoton microscopy with femtosecond laser-based metaphase plate ablation [72].

Cell fusion is used for nuclear transfer, hybridoma production, and reproductive and therapeutic cloning. Noncontact laser-assisted fusion is believed to be a promising tool for many biotechnological experiments on gametes and embryos. As close contact between cells is required for laser-assisted cell fusion, the blastomeres in two-cell embryos have usually been employed as model cells. Nevertheless, the fusion of oocyte and blastomere or MII-stage oocyte fusion is also possible [77,78] with ZP removed beforehand and lectin added for cell aggregation.

Hypothetical models explaining the process of femtosecond laser-induced cell fusion have been described by Gong et al. [82] and later by Katchinskiy [83]. The cell membrane consists of a phospholipid bilayer with hydrophobic tails buried in the interior of the membrane and the polar head groups (hydrophilic) project outwards. Exposure to femtosecond laser pulses causes rearrangement of the phospholipid membrane. Multiphoton and avalanche ionization (when the laser pulse intensity exceeds the threshold value) take place at the laser focal volume during the membrane exposure, which results in high density of ions and electrons destabilizing the phospholipid molecules. The ions and electrons can penetrate the membrane's exterior layer and cross over to the central nonpolar region and break the bonds between the fatty acid tails. As a result, the ionized molecules form new bonds with phospholipid molecules of the adjacent cell thus creating a single phospholipid bilayer. Not bonded phospholipid molecules that are too far apart from the adjacent cell do not cross-link and link back to form their original structure.

Karmenyan et al. [71] have demonstrated the successful fusion of two blastomeres at the two-cell stage by picosecond laser resulting in the generation of viable tetraploid mouse blastocysts. Cell fusion efficiency depended on the site of the laser action and reached 35.8% when laser affected the ends of the contact border of the blastomeres and 66.7% when the middle region of the blastomere contact border was exposed to laser radiation. Moreover, the number of fused embryos that developed to the blastocyst stage in the second group was substantially higher (95%) than in the first (33.3%) [71,76]. Blastocyst hatching occurred only in the control group, while none of the fused embryos hatched.

Kuetermeyer et al. [73] have depicted the formation of a long-lasting vapor bubble (4–8 $\mu$m in diameter) in the irradiated area, creating a pore in both adjacent cell membranes, a prerequisite for successful blastomere fusion. The dependence of gas vapor diameter and its lifetime upon laser pulse energy and the exposure time was studied [76], and possible toxicity of gas bubble formation was supposed. The fluence threshold for the fusion of

two-cell porcine embryos was estimated at $F$ = 180–280 mJ cm$^{-2}$. The maximum fusion efficiency increased with decreased exposure time [73].

Several studies have assessed the ploidy of fused embryos [73,76,78]. All metaphase plates exhibited a tetraploid karyotype post-laser-assisted fusion of mouse blastomeres [76], while diploid cell nuclei and tetraploid cell nuclei were identified after the fusion of two-cell porcine embryos [73]. Various mechanisms of tetraploidization post-laser-induced cell fusion have been supposed. Osychenko et al. [76] suggested that tetraploidization may have occurred due to the essential process of DNA duplication before the second mitotic division. However, a recent, detailed study [78] revealed that tetraploidy occurred through the mechanism of typical metaphase plate formation. Intracellular transport during the process of femtosecond laser-induced cell fusion was studied by cell labeling with fluorescent dyes, green fluorescent protein (GFP), and by the in vivo laser-induced generation and tracking of fluorescent particles [78].

Ilina et al. [74,75] have proposed using second harmonic radiation of femtosecond Cr: forsterite laser ($\lambda$ = 620 nm, $\tau$ = 100 fs, $f_{rep}$ = 10 Hz) for blastomere fusion. The irradiation of the central region of the plasma membrane in the interface between two blastomeres (Figure 7a) with a single laser pulse with an energy $E$ = 35 nJ (fluence $F$ = 0.5 J cm$^{-2}$) [74] resulted in successful cell fusion in 88.9% of cases. The fusion was completed within 20–60 min post-irradiation (Figure 7b–d), and about 50% of the fused embryos developed normally up to the blastocyst stage. Fluorescence staining with FM4-64 dye shows a single membrane surrounding the cytoplasm of two fused blastomeres (Figure 7e,f), while Hoechst 33258 staining demonstrates a hybrid cell with two nuclei (Figure 7g,h).

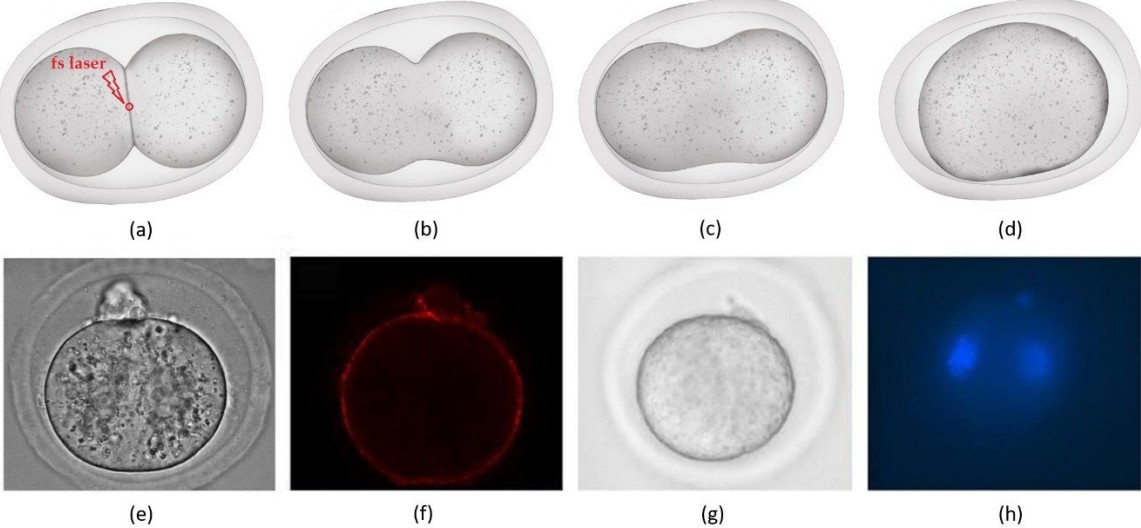

**Figure 7.** (**a**–**d**) Schematic representation of blastomere fusion under the action of femtosecond laser pulse, (**e**,**g**) brightfield images of mouse embryos after laser-induced blastomere fusion with (**f**) fluorescence image of hybrid cell membrane stained by FM4-64 dye and (**h**) nuclei stained by Hoechst 33258 dye (Ref. [74]).

## 5. Optoporation and Laser Transfection of Developing Embryos

Targeted delivery of biomolecules into cells in living organisms is one of the most crucial techniques in developmental biology. Microinjection and electroporation have been widely used for the cytoplasmic introduction of foreign materials into developing embryos. In recent years, substantial interest has occurred in developing alternative techniques for precisely delivering various molecules, including DNA, mRNA, or siRNA, into the blastomeres of embryos for cell-selective genetic modification. Developing a novel noncontact technique for introducing foreign molecules into chorionated embryos without disrupting the protective membrane is critical for developmental biology.

Kohli et al. [84,85] evaluated the applicability of femtosecond laser pulses as an alternative method for delivering exogenous material into the embryonic cells of zebrafish (*Danio rerio*) embryos (Table 2). Transient pores were created by focusing femtosecond laser pulses onto the individual blastomeres of embryos. The delivery of a fluorescent probe and fluorescein isothiocyanate (FITC) into the early cleavage- to early blastula-stage (2-cell to 128-cell) dechorionated embryos as well as into chorionated embryos (without disruption of the protective membrane) has been demonstrated. Moreover, the laser-based introduction of both exogenous streptavidin-conjugated quantum dots and simian CMV-EGFP plasmids into the blastomeres of the early-to-mid-cleavage stage (2-cell to 8/16-cell) dechorionated embryos has been illustrated. It should be noted that the survival of the laser-targeted chorionated and dechorionated embryos approached 100%. The optimal parameters for laser poration (optoporation) were defined [84,86] concerning both the minimum average laser power and beam dwell time visually, resulting in the ablation of blastomeres. Early-to-mid-cleavage stage (2-cell to 4/8-cell) zebrafish embryos were ablated using an average laser power $P_{av}$ of 25–50 mW and a beam dwell time of 5–500 ms. $P_{av}$ = 40–45 mW and a beam dwell time <100 ms were determined to be the ideal laser parameters for transient pore formation [86]. Researchers have also analyzed the key developmental features of laser-targeted embryos compared to control embryos to address whether femtosecond laser ablation induces any short- or long-term developmental effects in embryos. No significant differences in hatching rates and developmental morphologies were observed in laser-treated embryos relative to controls. Thus, such a technique represents an effective tool for laser-based noncontact and non-destructive delivery of exogenous material into living cells and may be helpful in developmental biology and other disciplines, including cryobiology and embryology.

**Table 2.** Femtosecond laser-based poration and transfection of embryos.

| Type of Embryos | Type of Manipulation | Laser Exposure Parameters | References |
|---|---|---|---|
| Zebrafish embryos | Optoinjection and transfection | Ti:sapphire laser<br>$\tau$ = sub-10 fs,<br>$\lambda$ = 800 nm,<br>$f_{rep}$ = 80 MHz,<br>$P_{av}$ = 40–220 mW,<br>$E$ = 0.5–3 nJ,<br>$I_{peak}$ = $10^{11}$–$10^{12}$ W cm$^{-2}$,<br>$P_{av}$ = 120–160 mW, dwell time: 200–500 ms<br>(for quantum dots introduction)<br>$P_{av}$ = 40–45 mW, dwell time: 200–500 ms<br>(for transfection) | [84,85] |
| *Pomatoceros lamarckii* embryos | Optoinjection | Ti:sapphire laser<br>$\tau$ = 180 fs,<br>$\lambda$ = 800 nm,<br>$f_{rep}$ = 80 MHz,<br>$P_{av}$ = 52–78 mW,<br>t = 10–40 ms | [87] |
| Zebrafish,<br>Chick,<br>Shark and Mouse embryos | Optoinjection and transfection | Ti:sapphire laser<br>$\tau$ = 120 fs,<br>$\lambda$ = 800 nm,<br>$f_{rep}$ = 1 kHz, 50 pulses for exposure<br>$E$ = 100–800 nJ | [88] |

Unlike Kohli et al., who described optoporation of relatively large (~1 mm) zebrafish embryos with NIR femtosecond laser pulses, Torres-Mapa et al. [87] performed optoporation for the intracellular delivery of a range of impermeable molecules into the blastomeres of the 60 μm-sized *Pomatoceros lamarckii* embryo (Table 2). Researchers have employed a holographic system based on a spatial light modulator that can be used for laser targeting individual blastomeres and stable embryo trapping. Optical trapping of embryos was achieved by switching to the continuous-wave mode of the Ti:sapphire laser employed for

optoinjection. The optoinjection of 3–500 kDa dextrans (fluorescently labeled with Texas Red and fluorescein) into the blastomeres of chorionated embryos (without the need to remove the outer membrane of the embryo) and the delivery of propidium iodide into the inner layers of cells in well-developed embryos have been demonstrated. The formation of a gas bubble was used as an indicator of membrane disruption leading to the rapid diffusion of the dye into the targeted blastomere. As large bubbles often led to the leakage of blastomere contents and compromised embryonic development, the size of the bubble needs to be well-controlled (5 μm in size was considered optimal) to retain normal embryonic development post optoporation. The laser parameters were set to $P_{av}$ = 65 mW and exposure time t = 30 ms. Optoinjection efficiency was evaluated as nearly 44% for single-cell zygotes and 55% for late-stage (>16 cells) embryos. The optoinjected blastomere remained viable, and the injected dye was passed on to daughter cells. Thus, the proposed technique may be used for cell lineage mapping at early and later stages of embryonic development and for genetic modification in transgenic animals.

Hosokawa et al. [88] demonstrated the efficiency of the optoporation technique using a femtosecond laser amplifier with high pulse energy (E > 100 nJ) and low repetition rate ($f_{rep}$ = 1 kHz). First, antisense morpholino oligonucleotides (1810 Da), dextran (10,000 Da), and DNA plasmids have been introduced into the cells of zebrafish and chick embryos (Figure 8). Then, DNA plasmids have been successfully delivered into single neurons of chick embryos. Finally, the fate of individual neurons in non-transgenic zebrafish embryos has been manipulated by the targeted introduction of mRNA. The photoporation experiments were performed with pulse energy 2 (100 nJ) to 16 (800 nJ) times higher than the threshold energy for cavitation bubble generation in water. Efficient delivery of FITC-tagged morpholinos into single cells of zebrafish embryos (28 h post fertilization (hpf)) was achieved when laser pulses with an energy of 300–400 nJ were applied (50 pulses at 1 kHz). The energy of 400 nJ was enough to introduce a relatively larger molecule, dextran, into a single cell of zebrafish embryos (25–26 hpf) with a success rate of 76.5%, while higher pulse energy (800 nJ) was required to introduce molecules into the cells of later-stage embryos (30 hpf) with a success rate of 54.5%. Successful transfer of dextran into single epithelial cells of chick embryos and DNA plasmid delivery into both external and internal cells (single neurons) was achieved by applying 400-nJ laser pulses. The suggested optoporation technique was also applied for delivering 10,000 Da dextran into single neurons of vertebrate embryos (E9 mouse embryos and stage 29 shark embryos).

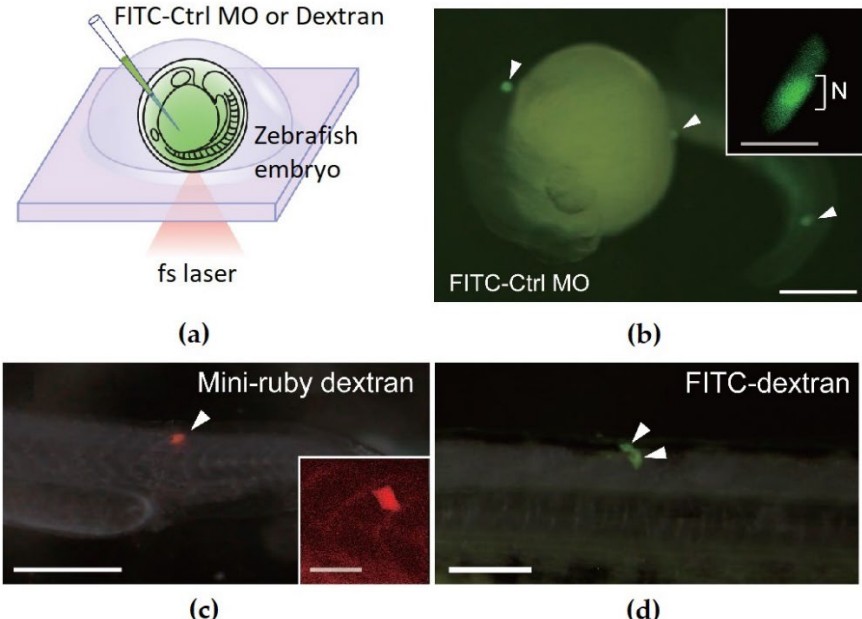

**Figure 8. (a)** Schematic representation of laser photoporation of morpholino oligonucleotides or dextran

into single cells of zebrafish embryos (FITC-MO, mini-ruby dextran, or FITC-dextran was injected into the chorion cavity of anesthetized embryos mounted in methylcellulose solution. The femtosecond laser pulse train was focused on the surface of single cells), (**b**) FITC-MO was delivered into three single cells (arrowheads) of 28-hpf zebrafish embryo and localized to the nucleus (N) of the targeted cells, (**c**) mini-ruby dextran was delivered into a single cell (arrowhead) of a 25- to 26-hpf zebrafish embryo and (**d**) FITC fluorescence was detected in newly divided cells (arrowheads) 24 h post FITC dextran optoinjection. Scale bars: 250 μm (**b**,**c**); 50 μm (insets in (**b**,**c**)); 100 μm (**d**). Adapted from open-access source, Ref. [88].

## 6. Ultrashort Laser Microsurgery of Externally Developing Embryos

ULPs are successfully applied for microsurgery of preimplantation mammalian embryos and microsurgery of various externally developing organisms like *Drosophila melanogaster* [89–92], zebrafish, sea urchin, and starfish embryos [93,94]. This section briefly reviews the results reported in these studies and summarizes them in Table 3.

**Table 3.** Femtosecond laser microsurgery of externally developing embryos and embryonic organs.

| Embryo/Organ Type | Type of Manipulation | Laser Exposure Parameters | References |
|---|---|---|---|
| Drosophila melanogaster, Oregon-R strain | 3D laser ablations | Ti:sapphire laser oscillator and an optical parametric oscillator<br>$\tau = 130$ fs,<br>$\lambda = 830$ nm (for ablation), 920 nm (for TPEF), 1180 nm (for THG)<br>$f_{rep} = 76$ MHz,<br>$P_{av} = 50$–275 mW,<br>$I_{peak} = 3 \times 10^{12} - 2 \times 10^{13}$ W cm$^{-2}$,<br>$E = 0.7$–3.6 nJ<br>$I_{peak} = 7 \times 10^{11}$ W cm$^{-2}$ (for TPEF)<br>Number of line scans–3, 100 μm long | [89] |
| Drosophila melanogaster embryo (stage 15) | 3D laser ablations | Frequency tripled Nd:YAG laser<br>$\tau = 470$ ps,<br>$\lambda = 355$ nm,<br>$f_{rep} = 1$ kHz, 100 pulses,<br>$E = 0.74 \pm 0.04$ μJ,<br>$I_{peak} = 744 \pm 40$ GW cm$^{-2}$ | [90] |
| Drosophila melanogaster embryo | Local tissue immobilization, actomyosin meshwork ablation | For tissue immobilization:<br>$\lambda = 1030$ nm,<br>$f_{rep} = 50$ MHz,<br>$\tau = 200$ fs<br>t = 40 ms,<br>$P_{av} = 200$ mW,<br>For actomyosin meshwork ablation<br>$\lambda = 950$ nm | [91,95] |
| Drosophila embryo Zebrafish embryo | Nuclei ablation Neuron dissection Tissue cauterization Optogenetic manipulations | $\lambda = 1025$ nm,<br>$f_{rep} = 54$ MHz,<br>$\tau = 180$–200 fs<br>For nuclei ablation<br>t = 15 ms, 4 times<br>$P_{av} = 800$ mW,<br>For soma ablation:<br>t = 9 ms, 5 times<br>$P_{av} = 880$ mW<br>For axon ablation:<br>t = 3 ms<br>$P_{av} = 604$ mW<br>For tissue cauterization:<br>t = 40 ms<br>$P_{av} = 400$ mW<br>Optogenetics:<br>$\lambda = 780$ nm<br>$f_{rep} = 80$ MHz,<br>$\tau = 150$ fs | [96] |

**Table 3.** *Cont.*

| Embryo/Organ Type | Type of Manipulation | Laser Exposure Parameters | References |
|---|---|---|---|
| Drosophila embryo | Epithelial tissue microsurgery | Ti:sapphire laser $\tau = 160$ fs, $\lambda = 870$ nm, $f_{rep} = 76$ MHz, $P_{av}$ (before the objective) = 20–40 mW (for ablation) $P_{av}$ (before the objective) = 8 mW (for imaging) $E = 0.13$–$0.15$ nJ (for microablation after dorsal closure) $E = 0.095$ nJ (for microablation during dorsal closure) $\upsilon = 0.66$ μm ms$^{-1}$ Number of line scans–160, 13 μm long | [92] |
| Sea urchin embryo, Starfish embryo, Zebrafish embryo | Dendrite ablation, mitotic pole ablation, plasma membrane and nuclear envelope wound | Ti:sapphire laser $\tau = 100$ fs, $\lambda = 800$ nm, $f_{rep} = 76$ MHz, $P_{av} = 60$ mW (at the specimen), $E = 0.8$ nJ | [93] |
| Zebrafish embryo | Ablation of mesodermal cells | $\tau = 300$ fs, $\lambda = 820$ nm, $f_{rep} = 80$ MHz | [94] |
| Zebrafish embryo | Opto-disruption of vascular structure | $\tau = 150$ fs, $\lambda = 800$ nm, $f_{rep} = 1$ kHz $F = 5$–$50$ J cm$^{-2}$ $I_{peak} = 30$–$300 \times 10^{12}$ W cm$^{-2}$ | [97] |
| Avian embryo heart | Ablation of the superior atrioventricular (AV) cushion and vitelline vessels | $\tau = 50$ fs, $f_{rep} = 1$ kHz, $E = 200$ nJ | [98,99] |

Supatto et al. [89] performed 3D microdissections inside live *Drosophila* embryos with ULPs to locally modify the structural integrity of embryos and thus modulate remote morphogenetic movements. The same laser source was used to conduct nonlinear microscopy (TPEF and THG) and track the changes after intravital ablations. The main advantage of femtosecond laser-induced ablation for embryo studies is that, due to the nonlinear dependence of photodestructive effects on excitation intensity, inner embryo structures can be processed while leaving the outer membranes intact. The authors explored the effects of average power, $P_{av}$, from 50 to 275 mW and a scan speed of $\upsilon = 0.2$–$50$ μm ms$^{-1}$. The laser power was attenuated to $P_{av} \sim 12$ mW to record a TPEF image of the induced fluorescence in the ablated area. The influence of microdissections on cellularization has been studied. The photodestruction of the embryos' particular region has been shown to cause rapid, long-range modulation of morphogenetic movements connected to the targeted area. Such an all-optical approach may find many applications in developmental biology, as it allows for studying the interplay between cell deformations and molecular signaling.

Morphogenetic movements during *Drosophila* embryogenesis can be also controlled by an optogenetic approach [100,101]. Optogenetics enables precise and noninvasive control of cellular activity by exploiting genetically encoded photo-activatable proteins or protein modules. Guglielmi et al. [100] applied optogenetic method to locally modulate cell contractility during tissue morphogenesis, and demonstrated the validity of the method in addressing the interplay between tissue geometry and force transmission during ventral furrow formation. The authors used optogenetic system based on the CRY2- CIB1 protein dimerization module that can be rapidly activated by blue-light illumination with no need for the addition of an exogenous chromophore. Single- and two-photon laser illumination with quantitative imaging were combined to control apical constriction with spatial and temporal precision. Two-photon illumination protocol allows precise spatial photo-activation with cellular resolution and is suitable for thick specimens as longer

wavelengths have higher penetration depth than shorter ones. Photo-activation with two-photon illumination was achieved using a femtosecond (140-fs) pulsed laser (Chameleon Ultra II; Coherent) at a repetition rate of 80 MHz, laser power of 3 mW (measured at 1 cm from the objective). The wavelength was set to 950 nm (475 nm) as it is the most effective wavelength in triggering the translocation of CRY2-OCRL from the cytosol to the plasma membrane. The translocation of CRY2-OCRL was shown to be correlated with the laser power used to trigger photo-activation. So, it is possible to modulate the extent, to which apical constriction is inhibited, by increasing or decreasing the laser power used to trigger CRY2-OCRL translocation to the plasma membrane. Izquierdo et al. [101] reported on reconstitution of epithelial folding, a conserved morphogenetic process driving internalization of tissues during animal development. An optogenetic system was used to activate Rho signaling at the apical surface of developing Drosophila embryos (stage 5) prior to any sign of morphological differentiation. Photo-activation using a two-photon based protocol at a wavelength 950 nm with laser power set to 10 mW caused fast recruitment of RhoGEF2-CRY2 to the plasma membrane, and resulted in the moving away of cells from the imaging plane in an area that precisely matched the geometry (circular, triangular, or squared) of the illuminated area. The induced furrows were shown to occur at any position along the dorsal–ventral or anterior–posterior embryo axis in response to the spatial pattern and level of optogenetic activation independently of any pre-determined condition or differentiation program associated with endogenous invagination processes.

Engelbrecht et al. [90] have combined plasma-induced laser nanosurgery with optically sectioning light-sheet-based fluorescence microscopy and applied them to various 3D biological model systems. Light-sheet-based microscopy is suitable for imaging fixed tissues and live samples with a high spatial resolution at subcellular levels with minimal phototoxicity. A single plane illumination microscope was used to study hemocyte migration following local laser injury of *Drosophila melanogaster* embryos. The indicated area was irradiated with 100 pulses ($\lambda$ = 355 nm, $\tau$ = 470 ps, $f_{rep} \leq$ 1 kHz, and $E$~0.74 $\mu$J). Following irradiation, GFP-labeled hemocytes moved to the "wounded" area, growing protrusions and potentially removing damaged cells and cellular debris. The outer shell of the embryo remained entirely intact during the whole procedure. The authors depicted that highly precise, noncontact, three-dimensionally confined plasma-induced laser ablation could be successfully applied in manipulating living specimens in 3D.

Laser microsurgery is a valuable tool in developmental biology to probe cell mechanics to understand how cells and tissues acquire and maintain their shape. During tissue morphogenesis, cells are equipped with networks of actomyosin-generating forces. A technique based on near-infrared fs pulsed laser ablation of actomyosin networks in developing *Drosophila* embryos while preserving the membranes' integrity was described in detail [95]. To demonstrate membrane integrity after laser ablation, fluorescein (0.9 kDa, UV uncageable) was injected in the embryo yolk during the stage of cellularization and then diffused in all cells of the embryo. The same infrared laser source was used for actomyosin network ablation and fluorescein uncage (in that case the laser power was ~10-fold less than the power used for ablation). The objective was focused in the center of one of the two cells sharing the ablated actomyosin junction. Fluorescence signal emitted from only one cell proved the membrane integrity, while fluorescent signal from both cells proved the membrane perforation.

The results of experimental investigation and numerical modeling of cell behavior during early gastrulation in *Drosophila* embryos was reported by Rauzi et al. in [91]. A custom-built system coupled to multiview selective plane illumination microscopy (MuVi-SPIM) was employed to perform a local tissue immobilization along a line of 150 $\mu$m by exposing the apical side of the epithelium, as close to the vitelline membrane as possible, to NIR femtosecond laser pulses ($\lambda$ = 1030 nm, $P_{av}$ = 200 mW, and $t$ = 40 ms). Actomyosin meshwork ablation was also performed using a femtosecond-pulsed infrared laser tuned at 950 nm. The authors demonstrated interdependencies of epithelial movements during the early stages of gastrulation. They revealed the correlation of changes in the behav-

ior and shape of different types of cells with their apical actomyosin architecture and biomechanical tension.

Medeiros et al. [96] coupled a MuVi-SPIM with an infrared femtosecond laser to target any region within an embryo and visualize the sample in 3D. They demonstrated this tool's several applications in two embryonic model systems. First, they performed laser ablation of individual nuclei in the early *Drosophila* embryo syncytium to investigate the mechanism that determines the spatial distribution of the nucleus. Selected nuclei were ablated ($\tau$ = 180 fs, $\lambda$ = 1025 nm, $f_{rep}$ = 54 MHz, $P_{av}$ = 800 mW, $t$ = 15 ms, and four exposures) before mitotic division ten to see whether damaged nuclei can reach the cortex after mitotic division nine. Most ablated nuclei failed to reach the cortex, while the neighboring nuclei moved to the cortex of the embryo. Then, the system was employed for selective soma and axon laser ablation (with an average power of 880 mW ($t$ = 9 ms, five exposures) and 604 mW ($t$ = 3 ms) correspondingly) and monitoring the response of microglia in the zebrafish embryos. A microglial cell reached the ablated soma 20 min post-ablation. Laser-injured axon was retracted, and the remaining part of the axon was dragged away by one microglia at 8.5 min post-ablation. The system was also used for tissue cauterization and monitoring eventual tissue flow perturbation at the embryo scale in the *Drosophila* embryos, as well as for selective activation of photo-sensitive proteins in 4D for perturbation of tissue morphogenesis while performing in toto imaging.

Thayil et al. [92] performed laser-based microsurgery of the epithelial tissue within a *Drosophila* embryo at the final stages of its embryonic development combined with TPEF imaging. They conducted ablation of GFP-labeled and unlabeled tissues during and after dorsal closure. The authors revealed the capability of MHz ultrashort pulse oscillators to introduce controlled wounds in embryos. The energy threshold for tissue ablation during dorsal closure decreased compared to post-closure tissue ablation. The pulse energy required for ablation at this stage was lower (~28%) than at later stages. The additional tensile forces present during closure probably caused this reduction. Moreover, the authors found that ablation of unlabeled tissue required higher energy deposition than GFP-labelled tissue, ensuring multiphoton-mediated ablation. Finally, they observed increased actin activity near the wound edges as the tissue tried to heal from the wounds. The presented results and further investigations are critical for understanding intercellular signaling events and the dynamics of tensile forces during wound healing.

Controlled damage has been introduced to thick specimens [93] employing a Ti:sapphire laser integrated within a commercial multiphoton microscope. Laser pulses of $\tau$ ~100 fs duration have been applied for severing a dendritic branch of Rohon-Beard neurons within zebrafish embryos, ablation of a mitotic pole within 2-cell-stage urchin embryos, and wounding of the plasma membrane and nuclear envelope within a starfish oocyte. The authors demonstrated the lack of toxicity of laser-initiated wounds for the surrounding cells or cytoplasm. The ability to sever neuronal processes at previously hard-to-reach locations provides an opportunity better to investigate axonal degeneration, regeneration, and pathfinding mechanisms. Moreover, in urchin embryos, laser-induced damage at the center of a mitotic aster was depicted to result in normal completion of the first division but second division failure, thus indicating that normal development of blastomeres was impossible without a centrosome. The cytoplasmic changes in post-laser wounding of the plasma membrane within starfish oocytes were shown to be dependent on the site where damage was introduced (at the yolk-containing end or clear end). Localized multiphoton wounding of the nuclear envelope was performed to study the requirements for nuclear compartmentation and translocation during meiosis and resulted in nuclear collapse, indicating that loss of the compartmentation barrier makes the structure unstable.

Supatto et al. have presented a simple strategy for probing microscopic fluid flow in vivo based on an all-optical procedure combining femtosecond laser ablation, fast confocal microscopy, and 3D particle tracking [94]. Instead of using tracer particles injected with a needle to investigate flow dynamics in vivo, they employed subcellular femtosecond laser ablation to generate fluorescent micro-debris seeding the flow. Fast confocal imaging and

3D particle tracking imaged and quantified the seeded flow. Successful ablations generated intense fluorescence in the targeted region, and subcellular micrometer-scale ablation in the zebrafish tailbud was achievable down to a depth of ~70 μm. They also investigated the 3D motion of the flow with fluorescent particles within the cavity. The particles exhibited a circular motion around the dorsoventral direction. The detailed description of cilia-driven fluid movement presented in this study is essential for unraveling the relationships between flow and signal transduction crucial for maintaining the asymmetry of the embryo.

Developing zebrafish embryos are also considered a functional model system to examine the mechanisms of blood vessel formation during development. Woo et al. [97] have applied ULPs ($\lambda$ = 800 nm, $\tau$ = 150 fs, and $f_{rep}$ = 1 kHz) to selectively induce damage to vascular-related structures in transgenic zebrafish embryos at different developmental stages. The threshold fluence for lesion formation of the vascular endothelium has strongly been shown to depend on the developmental stage of the embryos. In the early developmental stage, the vascular endothelium was very vulnerable to ultrafast laser irradiation compared to those in later developmental stages. While the disruption of the vascular structure for Somite 14, 20, and 25 stages occurred at the laser fluence $F$ in the range of 5–7 J cm$^{-2}$, the induction of apparent lesions of the blood vessels in the later development stages (Prim 16 and 20) were induced at the laser fluence level of 30–50 J cm$^{-2}$.

Yalcin et al. [98] developed a non-invasive optical technique for studying hemodynamic signaling in cardiac morphogenesis based on TPEF coupled with femtosecond pulsed laser ablation. ULPs were applied to create localized microscale defects within avian embryo hearts (a~100 μm spherical void in the superior atrioventricular cushion was created) that may mimic clinical congenital heart defects. TPEF was used to monitor cushion and simultaneously visualize changes in the hemodynamic environment inside embryo hearts post laser-induced tissue disruption. The proposed non-invasive technique may dramatically enhance the ability to dissect the complex and interrelated effects of genetic and hemodynamic signaling likely driving clinical congenital heart defects. Recently, the authors [99] used a multiphoton microscopy-guided femtosecond pulsed laser ablation system to study the vascular remodeling of embryonic blood vessels under altered blood flow environments. The blood flowing inside the vitelline vessels of chicken embryos was selectively ablated to induce clot formation inside these vessels. The clotting energy increased linearly with the diameter of the vessel. The blood flow in the irradiated vessel was blocked, causing immediate dramatic changes to the blood velocity in the upstream and downstream vessels. Vitelline vessels proved very sensitive to alterations in hemodynamic forces and remodeled in response to load changes.

## 7. Ultrashort Laser-Based Microsurgery and Microscopy of Preimplantation Embryos in Assisted Reproductive Technologies: A Prospects

Ultrashort lasers are believed to be a promising alternative to infrared 1.48 μm diode lasers with micro-to-millisecond pulse duration for application in assisted reproduction. The latter are widely used for oocyte or embryo microsurgery in in vitro fertilization (IVF) laboratories worldwide. Laser pulses are mainly applied to create an opening in the outer shell of the oocyte or embryo, called zona pellucida (ZP). Infrared diode lasers seem to be an effective and safe tool [102–104]. Nevertheless, strong recommendations regarding optimum regimes for embryo exposure, in particular, reducing the pulse lengths in laser sources intended for use in clinical practice or keeping safe distances between the laser firing position and the nearest blastomeres [105–108] should be considered to minimize possible laser-related thermal risks. Due to safety concerns, most ZP microsurgical procedures with milli-to-microsecond laser pulses are performed at the early stages of preimplantation embryo development (when sufficient perivitelline space exists between the ZP and embryonic cells). Novel solutions to issues in assisted reproduction based on the application of ULPs have been proposed recently to overcome this limitation and reduce the risk of embryo damage.

First, ULPs have been successfully applied for creating the hole in the ZP to perform oocyte or embryo biopsy (i.e., removal of one or several embryonic cells to perform preimplantation genetic diagnosis and screening for monogenic diseases or chromosomal abnormalities). Laser-based polar body biopsy as well as trophectoderm (TE) biopsy (at later stages of preimplantation development) can be performed in a fully noncontact manner by combining a femtosecond laser (for ZP microsurgery) and optical tweezers (for cell trapping and removal) in a single device [109–111].

Femtosecond laser pulses can be applied for partial ZP thinning or drilling to make it easier for the embryo to hatch out of its shell and facilitate embryo implantation. The procedure is called laser-assisted hatching (LAH). As ZP drilling at the late stages of preimplantation development may be more advantageous than ZP drilling at the early-stage embryo [112] (LAH with milli-to-microsecond laser pulses is usually performed at the cleavage stage embryos), the benefits of precise and delicate ZP microsurgery with femtosecond laser pulses have been employed [113] to perform ZP drilling at the late stage of preimplantation development to stimulate embryo hatching to start at a prescribed location. The procedure was called controlled laser-assisted hatching and demonstrated a high probability of embryo hatching through the artificial opening (93.3% through the hole created close to the TE and 97.4% through the hole created close to the inner cell mass (Figure 9)).

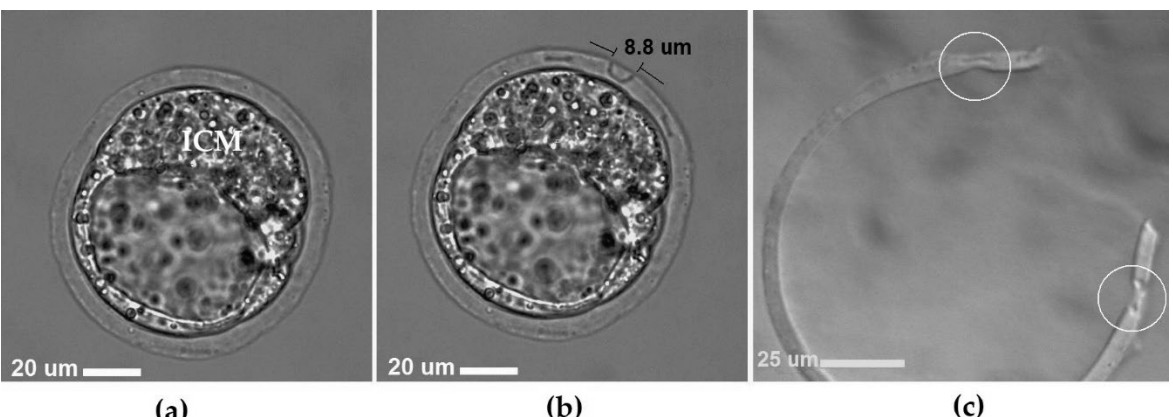

**Figure 9.** (**a**–**c**) Femtosecond laser-assisted ZP drilling close to the inner cell mass (ICM): (**a**,**b**) mouse embryo before and after ZP drilling. Artificial opening in the ZP and two additional incisions on either side of the opening are formed by femtosecond laser microsurgery, (**c**)—ZP after embryo hatching with clearly visible incisions (white circles) proving the fact that the embryo hatched right through the artificially created opening.

Moreover, femtosecond laser pulses have successfully been applied for individual embryo labeling [114]. The technology may be helpful in developmental biology for studying the characteristics of developing embryos during their co-culture in groups and in assisted reproductive technologies for preventing medical accidents related to mixing gametes between patients. Although such errors are rare, they periodically occur in IVF laboratories worldwide [115,116] and cause serious legal consequences and prolonged emotional distress for the patients. Femtosecond laser pulses (E~20 nJ, $\tau$ = 280 fs, $\lambda$ = 514 nm—the second harmonic of ytterbium laser, $f_{rep}$ = 2.5 kHz) were applied to precisely engrave alphanumeric codes (typically comprising 4–5 characters) in the volume of ZP of mouse embryos at the zygote stage to perform embryo labeling. Figure 10 illustrates codes "OLIV" and "VIVO" engraved on the ZP of mouse embryos (0.5 dpc) and the same embryos at 1.5 dpc and 3.5 dpc respectively. The engraved codes were recognized until ZP thinning before hatching 4.5 days post coitum, enabling embryo identification for nearly the entire period of preimplantation development [117]. Moreover, no differences in morphology and developmental rates in laser-labeled embryos compared to intact control embryos were observed.

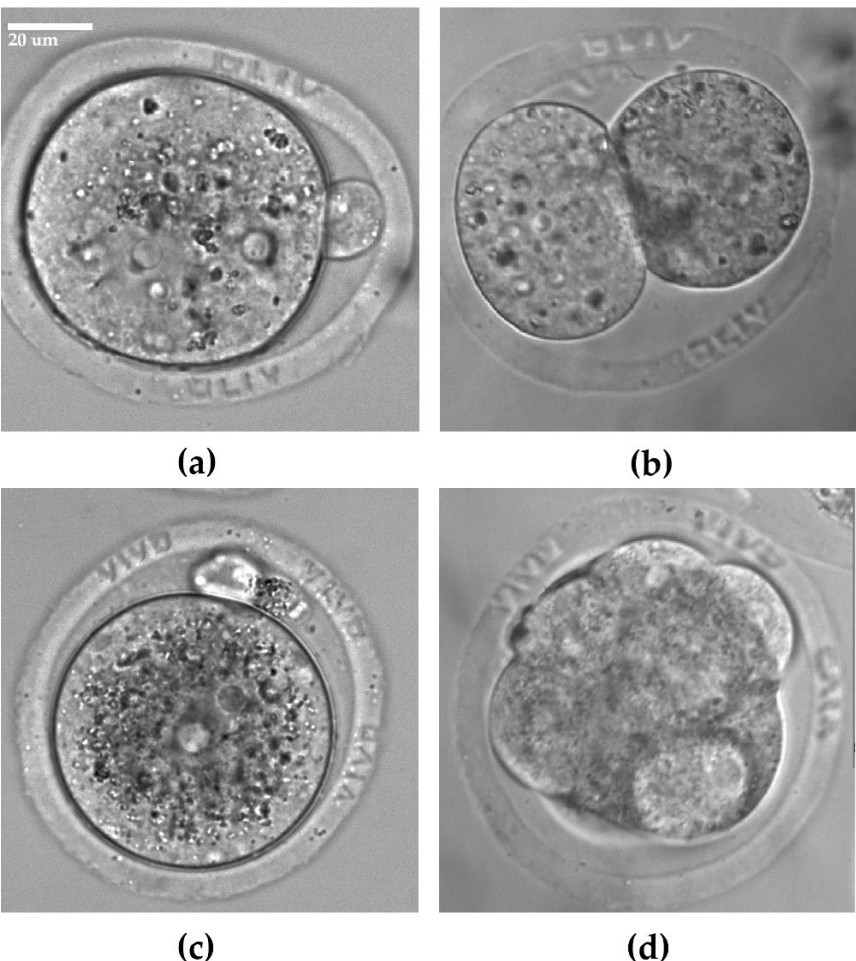

**Figure 10.** Femtosecond laser-assisted engraving of various codes on the ZP of preimplantation mouse embryos: (**a**) Embryo (0.5 dpc) right after the engraving of two codes "OLIV" and (**b**) embryo on the following day (1.5 dpc) with clearly visible codes on the ZP; (**c**) embryo (0.5 dpc) right after the engraving of three codes "VIVO" and (**d**) embryo 3.5 dpc with codes "VIVO" slightly out-of-focus but still recognizable.

As femtosecond lasers offer several vital advantages like high precision, minimal invasiveness, low collateral damage, and versatility over conventional milli-/microsecond lasers, it is expected that further advances in ultrashort laser technology aimed at reducing the complexity, size, and high cost of femtosecond lasers will help them gain popularity in the assisted reproduction.

Not only ultrashort laser-based microsurgery, but also nonlinear optical microscopy may become an indispensable tool in ART. It is known, that the selection of healthy embryos is important for increasing implantation potential in IVF. However, conventional imaging techniques have some drawbacks that should be taken into account. Confocal microscopy requires dye staining and is too invasive for IVF applications. Squirrell et al. demonstrated that conventional laser scanning confocal microscopy (laser intensity $I = 9 \times 10^3$ W cm$^{-2}$; 8 ms dwell time) inhibited the development of hamster embryos [118]. The authors suggested that the developmental arrest was caused by the generation of free radicals from the excited fluorophore, which damaged cellular components. Polarized light microscopy is a non-invasive tool to analyze human oocytes that allows the detection of anisotropic cell structures like meiotic spindle microtubules and zona pellucida glycoproteins [119]. However, this technique cannot visualize organelles and is insensitive to the detailed inner structures of embryos. With the development of novel microscopy techniques, much attention has been paid to the study of their safety and impact on the object under

investigation. Harmonic generation microscopes (SHG and THG) are capable of providing three-dimensional, label-free images of biological specimens with minimal phototoxic effects as compared to other imaging methods [120]. Thayil et al. confirmed the relative safety of embryo imaging with near-infrared femtosecond laser pulses and demonstrated the possibility of visualizing various embryo structures at different developmental stages with harmonic generation techniques [120]. The normal development of mouse embryos was demonstrated during 1-day-long discontinuous imaging ($\lambda$ = 1230 nm, $\tau$ = 65 fs, $f_{rep}$ = 76 MHz, $P$ = 35 mW), revealing the processes of morula compaction and blastocyst formation. SHG images demonstrated central spindles during cytokinesis, and THG images indicated lipid droplets, nucleoli, and plasma membranes. Hsieh et al. also used 1230-nm laser radiation for continuous imaging of mouse embryos for ten minutes with SHG and THG microscopy techniques [121]. The developmental status of the embryos was observed with a high three-dimensional spatial resolution, including the thinning of the ZP, expression of cell adhesion proteins, and cleavage of cells. The authors claimed that THG could provide the contrast required for cell membranes and laminated organelles, including the Golgi apparatus, endoplasmic reticulum, and mitochondria. Blastomeres, the nucleus, and the polar body were also clearly visible. SHG imaging enabled estimation of the thickness of the three layers of the ZP, which is an important factor in selecting proper oocytes. Kyvelidou et al. used THG imaging ($\lambda$ = 1028 nm, $\tau$ = 200 fs, and $f_{rep}$ = 50 MHz) to study blastomere equivalence [122]. The average laser power on the specimen was 20 mW (pulse energy $E$ = 0.4 nJ). It took 30 scans to obtain a 2D slice with high signal-to-noise ratio and 10–15 min for three-dimensional THG reconstruction. The authors observed an energy divergence of 12% to 18%, which could be valuable complementary information for selecting a proper blastomere for preimplantation genetic diagnostics and demonstrated that THG imaging of mitochondrial/lipid body structures could provide sufficient data on the energetic status of preimplantation embryos, time evolution of the different developmental stages, and embryo polarization prior to mitotic division.

Sanchez et al. [123] and Ma et al. [124] have also applied high harmonic generation microscopy techniques for embryo quality assessment. The ZP and meiotic spindle are the only subcellular structures in mammalian oocytes that produce SHG, with the spindle generating by far the largest signal [123]. Sanchez et al. employed a Ti:sapphire laser ($\tau$ = 150 fs, $f_{rep}$ = 80 MHz) with power values up to 80 mW (on the sample) to obtain high-quality images of spindles and reported that the SHG technique did not significantly impair embryo viability and could be a feasible and safe approach for non-invasive embryo assessment. Ma et al. applied THG microscopy combined with phasor fluorescence lifetime imaging microscopy (FLIM) [124] to capture endogenous fluorescent biomarkers of preimplantation embryos for quantitative identification of healthy embryos and prediction of their viability. It was shown that cleavage-stage embryos had a large amount of small, densely packed lipid droplets, whereas post-cleavage-stage embryos had large lipid droplets of low density. Dramatic changes in both lipid oxidation and lipid volume size started after the compaction stages.

Thus, considering the fact that there is no clinically approved staining dye for embryo and oocyte selection in IVF, it can be suggested that SHG and THG microscopy modalities might provide useful information for high-quality oocyte/embryo selection without compromising their viability.

## 8. Safety Aspects for Use of Ultrashort Laser Pulses

Embryos were shown to be extremely sensitive to environmental effects, including light (see comprehensive review for details [125]). Both the brightness and the wavelength of radiation can impact embryo development. Certain wavelengths of visible radiation can lead to an increase in heat shock protein (Hsp70) expression, generation of reactive oxygen species, and apoptosis [126]. Reactive oxygen species (ROS), including superoxide anions ($O_2^-$), hydrogen peroxide ($H_2O_2$), and hydroxyl radicals ($HO^-$), are highly reactive molecules. Intracellular ROS synthesis is regulated by various hormones, cytokines, and

growth factors. Under physiological conditions, cells have different mechanisms to cope with ROS production. An increase in the ROS levels above a certain threshold (so-called oxidative stress) is accompanied by processes like lipid peroxidation and oxidative modification of proteins and nucleic acids that are harmful for cell survival [127]. In embryos, oxidative stress may change the expression of important transcription factors involved in multiple cell biology pathways, and thus can negatively affect embryo development, quality, and viability [128]. So, it is very important to identify threshold concentrations of ROS that may negatively influence embryonic development [129].

Laser radiation is characterized by much higher intensity values than the light of incoherent sources (lamps), which cause photo-damage due to direct light absorption via single-photon transitions (especially UV and visible wavelengths). So, one should keep in mind that reckless use of a high-power laser radiation for cell manipulation can also results in photochemical damage to cells due to the generation of ROS and free radicals. These species are induced by excitation of absorbers, such as porphyrins, flavins, and coenzymes. When excess laser irradiation triggers ROS overproduction, not only cell components, such as proteins, lipids and DNA, begin to oxidize [130], but also redox homeostasis and cell cycle are disrupted [131]. The process of ROS generation under the action of femtosecond-laser radiation was described in details by Yan et al. [132]. The authors demonstrated that the ROS generation started solely in mitochondria and then ROS release to cytosol took place if the laser power exceeded a threshold value, thus indicating the balance between photodamage and cellular repair system was broken.

Femtosecond laser-induced ROS generation during the process of laser-based oocyte enucleation has been studied by Osychenko et al. [80]. Noncontact metaphase plate destruction was performed with 100 fs laser pulses at a wavelength of 795 nm ($f_{rep}$ = 80 MHz, $E$ = 0.5 nJ ($P_{av}$ = 40 mW), exposure time t = 60 ms). No statistically significant difference has been obtained between fluorescence level of H2DCFDA (used as an indicator for reactive oxygen species) in laser-exposed and negative control oocytes. The authors concluded that femtosecond laser enucleation could be an appropriate and rather safe method (for the given exposure parameters) for the recipient cytoplast preparation.

The repetition rate dependency of ROS generation during the process of laser-based cell microsurgery or nonlinear optical microscopy should also be taken into account. Baumgart et al. [133] considered the effect of laser pulse repetition rate (typically, kHz or MHz frequency range) on ROS generation during the cell nucleus irradiation with femtosecond laser pulses. Bovine endothelial cells were exposed to a series of pulses at a wavelength of 800 nm with energies $E$ of 1 and 1.5 nJ at 40 kHz and 4 MHz respectively. Significant increase of ROS concentration directly after laser manipulation followed by a decrease in both regimes at kHz and MHz repetition rate was observed. The influence of consecutive application of pulse trains from different repetition rate regimes was also studied. Irradiation with a MHz pulse train followed by a kHz pulse train resulted in a significantly higher increase of ROS concentration than in the reversed case and often caused cell death. The authors assumed that the ROS scavenger mechanisms were damaged or reduced in activity due to the first exposure. Therefore, addition of antioxidants during fs laser–based cell surgery experiments could be advantageous in terms of suppressing photochemical cell damage. In their later study, Baumgart et al. [134] focused on induced side effects during and after fs laser-based optoperforation for cell transfection. It was found that the uptake of extracellular $Ca^{2+}$ also strongly depended on the repetition rate and the irradiation time of the laser pulses, and the kHz-effects at the molecular level seemed to be reduced when compared to MHz pulses.

Even though the nonlinear optical microscopy was shown to be a powerful label-free imaging technology, one should avoid sample photodamage that may be induced by high-power ultrashort laser pulses. A comprehensive experimental study was performed by Talone et al. [130] to characterize the damage induced by focused femtosecond near-infrared laser pulses as a function of laser power, scanning speed and exposure time, in both wide-field and point-scanning illumination configurations. The data-driven approach provides

a predictive model that estimates damage probability and a safety limit for the working conditions in nonlinear optical microscopy. For example, the authors have demonstrated that cells can withstand high temperatures for a short amount of time, while they die if exposed for longer periods of time to even mild temperatures.

When lasers were first proposed for embryo manipulation in ART, certain concerns about their safety have been expressed [135]. Strong recommendations regarding exposure regimens have been proposed to minimize the risk of thermal damage when using commercially available infrared lasers (of micro-to-millisecond pulse durations) in clinical practice [107]. Hartshorne et al. [103] evaluated the levels of heat-shock proteins produced in embryonic cells after infrared diode laser treatment and demonstrated no increase in the level of heat-shock protein produced in nearby cells when zona pellucida drilling was performed. Honguntikar et al. [136] analyzed possible epigenetic changes in preimplantation embryos subjected to 1480-nm laser and showed that epigenetic signature in embryos were not significantly impaired by laser-assisted hatching. As safety is a priority when a new technique is introduced for human use in clinic, thorough analysis of possible negative effects of ULPs is a prerequisite for their successful implementation. Not only numerical modelling of possible thermal effects should be done [137], but additional experimental studies of possible long-term effects of ultrashort laser pulses on embryo development are to be performed.

## 9. Conclusions

Advances in ultrashort laser sources have facilitated their application in life sciences, and developmental biology is no exception. Due to nonlinear mechanisms of absorption of ULPs, energy deposition occurs in a small diffraction-limited volume in the bulk of a transparent material, thus minimizing possible damage to surrounding tissues. Therefore, the steady and continuous growth of applications of femtosecond or picosecond lasers for precise microsurgery and modification of cells and subcellular structures within living embryos has existed. Successful microsurgery and ablation of preimplantation mammalian embryos and externally developing embryos have been demonstrated. Moreover, femtosecond lasers are a safe, reliable, and efficient tool for precise three-dimensional imaging of embryos. NLO microscopy with femtosecond laser pulses includes TPEF, generation of the second and third harmonics, and CRS microscopy and offers several advantages over conventional fluorescence microscopy. NLO microscopy techniques are widely used in developmental biology for high-precision visualization of embryos, evaluating their quality and developmental potential, and investigating embryo change post-femtosecond laser-based microsurgery. It should be mentioned that cell microsurgery and imaging can be performed by using the very same femtosecond laser system. This review presents a few representative examples of simultaneous application of femtosecond lasers for ablation and imaging. While a boost in the application of ultrashort laser pulses in developmental biology has occurred over the past decades, we believe that further advances in ultrashort laser technology will result in the widespread use of ultrashort lasers in fundamental research and clinical practice (e.g., in the assisted reproduction). These advances include improving the reliability of laser operation, and easiness of use as well as reducing the size and high cost of these lasers. At the same time, one should always keep in mind that careful choice of laser parameters is required to avoid possible negative effects on embryo development during the microscopy or microsurgery procedure. Despite the numerous attempts of researches to create a model describing the mechanisms of cellular regulation, intracellular processes induced by ultrashort laser impact are not fully understood yet. Possible long-term effects of ULPs on embryo development that are virtually undetectable until the organism reaches late developmental stages and not covered in typical proof-of-concept publications should be a subject of extensive study.

**Author Contributions:** Conceptualization, I.V.I.; writing—original draft preparation, I.V.I. and D.S.S.; writing—review, editing, and visualization, I.V.I. and D.S.S.; funding acquisition, I.V.I. All authors have read and agreed to the published version of the manuscript.

**Funding:** This work was funded by the Russian Science Foundation, grant no. 22-24-00608.

**Institutional Review Board Statement:** Not applicable.

**Informed Consent Statement:** Not applicable.

**Data Availability Statement:** Not applicable.

**Acknowledgments:** This work was partially conducted using Unique Facility "Terawatt Femtosecond Laser Complex" in the Center for Collective Usage "Femtosecond Laser Complex" of JIHT RAS.

**Conflicts of Interest:** The authors declare no conflict of interest. The funders had no role in the design of the study; in the collection, analyses, or interpretation of data; in the writing of the manuscript; or in the decision to publish the results.

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
