# Peer review of "Application of Ultrashort Lasers in Developmental Biology: A Review"

_photonics, doi:10.3390/photonics9120914_

Round 1
Reviewer 1 Report
The review of I. V. Ilina and S. S. Sitnikov provides an overview on the recent advances in applying ultrashort LASER pulses in the field of developmental biology. Ultrashort pulses are indeed a versatile tool that can be applied to interact with the living in many ways including plasma based ablation, two-photon optogenetics and two-photon imaging. While the review is of interest, several points need to be addressed before further consideration for eventual publication.
11) Pag.2, the authors give the definition of an ultrashort pulse (1ps – 1fs). It should be clarified in the text that pulses below 100-80 fs are not appropriate for experimentation since the light is then absorbed by the optical elements used to direct the laser on the sample.
22) Pag.2, “One should clarify the peculiarities of their…”: unclear sentence. Rephrase.
33) Pag.7, The authors indicate to pulse duration at the LASER and sample level. It would be interesting to comment on the fact that tunable chirp systems are more often used to compensate for pulse widening. This results in pulse width conservation from the laser to the sample.
44) Pag. 7, the authors should better describe the process of cell fusion (i.e., local membrane breakage and inter-cell membrane fusion).
55) Pag.11, the authors should expand the comparison of 1ph-SPIM with 2-ph SPIM and include the corresponding 2-ph optical scheme.
66) The authors start with laser manipulation, then jump to laser imaging and go back to laser manipulation. This makes the text quite fractured. I strongly suggest to put the imaging part at the beginning or at the end of the manuscript.
77) Pag.17. Ref. [93] is a book presenting a very large spectrum of techniques that are also laser unrelated. The authors should refer specifically to the chapter of interest.
88) Authors could also include 3-photon activation of caged fluorophores (in place of 1-ph UV laser) to monitor cell integrity after ablation of cell-cell interfaces (Rauzi & Lenne 2015).
99) Comparison could be made between the fusion experiments and the junctional cortex ablation experiments with membrane preservation.
110) Pag.17, the authors should also refer to the paper Guglielmi et al. 2015 and Itzquierdo et al. 2018 (De Renzis lab) for the 2-ph optogenetic techniques (“activation of photo-sensitive proteins”) on confocal microscopy for synthetic morphogenesis.
Reviewer 2 Report
This review by Ilina and Sitnikov summarizes the applications in ultrashort lasers in developmental biology such as microsurgery and optical transfection among other potential uses of pulsed lasers.
The manuscript is well written and concise. It describes several applications in detail and reports the success rate of those applicatons. The manuscript successfully bridges the photophysical properties of pulsed lasers and how those properties can be applied for precision manipulation of embryonic development.
I have two minor criticisms for the paper:
-Authors description of SHG is highly confusing and should be rewritten using common phrases that were established in the previous SHG imaging review articles. For instance, authors say SHG is produced by a nonlinear material and give biological tissue as an example. Not all biological tissue structures would give SHG and not all the components of a biological tissue are ordered noncentrosymetric structures. Also, authors second sentence in the same paragraph regarding "medium should be noncentrosymetric at the excitation wavelength to obtain SHG signal" is highly misleading. Noncentrosymetry is a function of geometry and ordering of the atoms. If a material is noncentrosymetric, it is noncentrosymetric in all wavelengths. It does not lose this property based on incident wavelength. Yet, it is true that some materials have different SHG production efficiencies at different wavelengths which are related to their energy bandgaps, not their noncentrosymetry. These materials still produce SHG in all wavelengths, but some SHG signals are less intense and may appear dim in the detectors.
-Authors did an important job in categorizing all the applications and their potentials, but there is very little discussion on how underlying biology of embryonic development changes. I read the authors reporting embryo development rates with different laser exposures, but that is only a limited and extreme outcome. Given these applications are delivering high energy pulses to localized regions, they can induce activation of heat shock proteins and reactive oxygen species which can have long-term effects in the development that are virtually undetectable until the organism reaches late developmental stages that are not covered in typical proof-of-concept publications. I recommend authors to add a small section or a paragraph as a precaution that we are not fully aware of long-term effects of laser exposure on the developing embryo.They can also describe how researchers in the field should look into biology of laser exposure and see if the reactive oxygen species and other biological responses lead to changes in gene expression or cellular differentiation.
